# Reasoning about Ambiguous Definite Descriptions

**Stefan F. Schouten** and **Peter Bloem** and **Ilia Markov** and **Piek Vossen**

Vrije Universiteit Amsterdam

{s.f.schouten,p.bloem,i.markov,p.t.j.m.vossen}@vu.nl

## Abstract

Natural language reasoning plays an increasingly important role in improving language models' ability to solve complex language understanding tasks. An interesting use case for reasoning is the resolution of context-dependent ambiguity. But no resources exist to evaluate how well Large Language Models can use explicit reasoning to resolve ambiguity in language. We propose to use ambiguous definite descriptions for this purpose and create and publish the first benchmark dataset consisting of such phrases. Our method includes all information required to resolve the ambiguity in the prompt, which means a model does not require anything but reasoning to do well. We find this to be a challenging task for recent LLMs. Code and data available at: https://github.com/sfschouten/exploiting-ambiguity

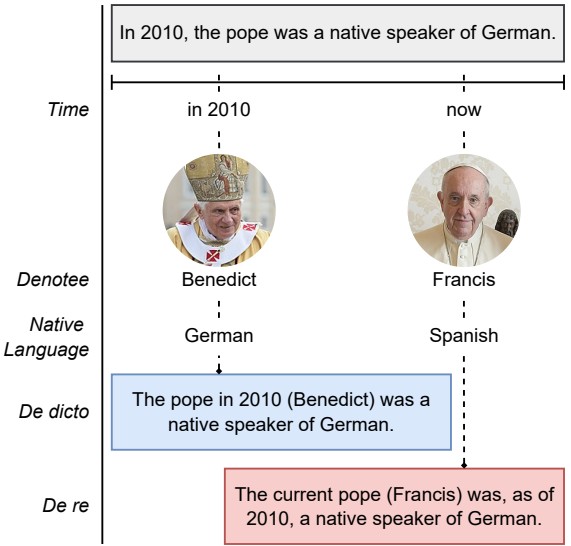

Figure 1: Example ambiguous definite description. Since a person's native language does not change over time, we know that the *de dicto* interpretation is correct.

## 1 Introduction

Natural language understanding and reasoning are interdependent skills: reasoning with natural language presupposes a level of understanding; but full understanding may require the resolution of ambiguities through reasoning. Complex ambiguity in particular could benefit from explicit 'out loud' reasoning, such as the reasoning that is produced with chain-of-thought prompts.

Existing resources used to evaluate reasoning are not well suited to investigate the capability of resolving ambiguity by explicit reasoning. Some existing benchmarks require the resolution of ambiguity, but focus only on ambiguity that humans can resolve intuitively (e.g. Winograd schemas, Levesque et al. 2012). The ability to reason with natural language is often evaluated with tasks considered complex enough to require it. These may or may not include ambiguities of various types, making them poorly suited to evaluate when models are able to resolve ambiguity and which types. Such tasks may also benefit from abilities besides explicit reasoning, such as factual recall. Thus, improvements on these tasks cannot be easily attributed to improvements in reasoning.

In this paper we create a new benchmark dataset which requires models to resolve ambiguous definite descriptions. Definite descriptions are phrases that denote entities by describing the properties or roles that are unique to them within the relevant context (e.g. "the pope", "john's mother", "our king"). We use ambiguous descriptions which denote one of two entities, and include information on both entities in context. Specifically, we introduce a *de dicto* / *de re* ambiguity by including a temporal operator (see Figure 1 for an example). By asserting something that is true of only one of these entities, one of the two interpretations can be excluded by reasoning.

We demonstrate the value of this approach by creating a new benchmark dataset generated from Wikidata. We explicitly include the knowledge required for disambiguation in the prompt of each

instance. Doing so ensures that when a system answers incorrectly, its failure stemmed from its inability to adequately reflect on the given information. We perform experiments showing the performance of recent large language models on our benchmark. We find that: (1) GPT4 and chain-of-thought prompting perform best (2) the LLaMA-based OpenAssistant model beats GPT3.5 in average accuracy, although the latter is more consistent; and (3) *de re* instances are harder than *de dicto* instances for all models.

In summary, our contributions are: (a) an outline of ambiguous definite descriptions, an under-explored class of problems useful for the evaluation of systems that reason with and about natural language (section 3); (b) a new benchmark dataset consisting of ambiguous definite descriptions, showcasing the value of the problem class (section 4), and finally (c) experimental results showing the performance of recent large language models on this benchmark (section 5)

## 2  Related Work

Reasoning is a broad term and the tasks related to it are varied. We focus on the type of reasoning that is explicit and is done with natural language, which has recently been surveyed extensively (Qiao et al., 2023; Huang and Chang, 2023; Yu et al., 2023). This paradigm became possible with the advent of large language models (LLMs) such as ChatGPT and GPT4 (OpenAI, 2023). These models can be prompted to perform 'chain-of-thought' reasoning (Wei et al., 2022); even in a zero-shot setting (Suzgun et al., 2022). This has significantly improved performance for many tasks (Suzgun et al., 2022). Using a new dataset of ambiguous definite descriptions, our experiments evaluate to what extent LLMs can use (chain-of-thought) reasoning to resolve ambiguity in language.

Recent work on ambiguity includes the construction of both curated (Liu et al., 2023; Min et al., 2020) and synthetic datasets (Yuan et al., 2023). Such datasets investigate ambiguity in a variety of tasks such as: natural language inference, open-domain question answering, etc. Our dataset is synthetically generated (but factual), and focuses specifically on *de dicto* / *de re* ambiguities, forming a binary classification task.

The term 'logical reasoning over natural language' (LRNL) was coined by Yang et al. (2023) to talk about a new trend where natural language

is used to represent knowledge and LLMs are used to reason over that knowledge. One challenge this paradigm could have to overcome is the potential for ambiguity inherent in natural language. Our work intentionally introduces ambiguity, and evaluates how well LLMs can resolve it by reasoning.

Previously, Winograd schemas (Levesque et al., 2012) have been used in various benchmarks to test for commonsense reasoning. These schemas consist of sentences with ambiguous pronouns which must be resolved in one of two ways. However, existing schemas focus on ambiguities that humans resolve intuitively (without the need for explicit reasoning). Creating new schemas that do require explicit reasoning does not appear straightforward, especially since creating them has proved difficult in general (Kocijan et al., 2023). Our method also involves ambiguous noun phrases, but requires explicit reasoning about definite descriptions rather than implicit reasoning about pronouns.

## 3  Ambiguous Definite Descriptions

Definite descriptions are noun phrases that denote entities by describing the properties or roles that are unique to them within the relevant context. Examples of such phrases in English include among others: "the pope", "john's mother", and "our king".

### De dicto vs. de re

The kind of ambiguities we use are known as *de dicto* / *de re* ambiguities. They involve a statement that is either true of what is said (*de dicto*) or true of the thing (*de re*). Take for example the sentence depicted in Figure 1, where it is unclear if the description "the pope" denotes the current pope (Francis) or the previous pope (Benedict). The source of the ambiguity is the phrase "In 2010", which can be read as primarily relating to the description ("the pope"), meaning the property is ascribed to the pope from 2010 (Benedict). Or, it can be read as relating primarily to the property ascription itself ("was a native speaker of German"), in which case the property is being ascribed to the current pope but is qualified as being true in 2010.

This type of ambiguity is the result of descriptions being combined with a non-extensional operator of which the temporal 'In PERIOD' is just one example. Other non-extensional operators could be used, such as modal operators, or propositional attitudes like 'PERSON believes that' or 'PERSON hopes that'.

| | Component | Explanation | Example |
|---|---|---|---|
| **Premises** | entity
-relations | A number of relations of the entity and the context in which it is true. | *Lars Lervik was part of the Telemark Battalion from January 2010 to January 2013.*
*Lars Lervik was part of the Brigade Nord from January 2018 to January 2020.* |
| | relation
-properties | The value of the property for each relation. | *Brigade Nord is a military unit of size class brigade.*
*Telemark Battalion is a military unit of size class battalion.* |
| | regularity | The relevant rule or regularity that is required to resolve the ambiguity. | *Military units do not change size class.* |
| **Question** | context | Establishes the current context. | **If the current date is** *June 2019*, |
| | sentence | An ambiguous sentence with a temporal operator and a property being ascribed to an entity denoted by a definite description. | **what is the most likely interpretation of the following sentence:** *The military unit of Lars Lervik was of size class brigade in March 2010.* |
| | interpretations | Two interpretations for the ambiguous sentence, one of which contradicts the regularity. | *1. Lars Lervik's unit in March 2010 (Telemark Battalion) was of size class brigade* ✗
*2. Lars Lervik's current unit (Brigade Nord) was of size class brigade in March 2010* ✓ |

Table 1: An example from the RADD-Wikidata-5-EN dataset (property pair P7779_Q21506450). The parts of the example that are bold are the same for each instance. For this example instance the *de re* interpretation is correct.

## 4 RADD-Wikidata-5-EN

To demonstrate the value of **R**easoning about **A**mbiguous **D**efinite **D**escriptions we create a (semi-)automatically generated dataset based on **Wikidata**. This dataset is based on **5** Wikidata property-pairs, and is in **En**glish. We focus on creating ambiguous sentences with a temporal operator. Wikidata contains many triples qualified with the 'start time' and 'end time' properties to indicate the period in which they were true.

In Table 1, one can see a complete example instance from this dataset. For a given main entity (e.g., Lars Lervik) we find the relations corresponding to that entity (Lars Lervik's military units and the beginning and end of his term with them). This information is combined with a property of the relations that does not change over time (the size class of the military units). Pairs of these relations are sampled such that the property (the size class) has different values for the two relations (in this case, one brigade and one battalion). Note that, by changing the property we ascribe to the denotee, we can flip which interpretation is correct (*battalion* instead of *brigade*) so we can generate two mirror instances for each entity and relation-pair.

We prioritized finding a small diverse set of property-pairs. Besides requiring a property whose value changes over time and a property whose value does not, the combination of properties also must yield sufficient results on wikidata (we required at least 50 to create the 100 samples). We also filtered out property-pairs where the results were dominated by a single entity, for example, had the data for the 'person-military_unit-size_class' pair consisted of only a handful of people changing units many different times, we would not have included it. See Table 2 for details on which property pairs we include and what they represent.

We include all facts relevant to the main entity in each prompt whether they are necessary to resolve the ambiguity or not, this allows us to measure if models can work around distractor facts.

### 4.1 Template design considerations

We use templates to generate the instances in the dataset, and while experimenting we identified a few things to avoid when phrasing them.

**Avoiding disambiguation by verb tense.** The ambiguous description must not contain a verb whose tense favours one interpretation over the other. For example with:

> *"In* MONTHYEAR, *the club for which* X *is/was head coach was from* Y."

If we use 'X **is** head coach' it implies that 'In MONTHYEAR' does not scope over the definite description, whereas with 'X **was** head coach' the implication is that it does scope over the definite description (assuming MONTHYEAR is in the past). Thus, we avoid using such phrases, for example, by formulating the example as:

> *"In* MONTHYEAR, *the club with* X ***as its** head coach was from* Y."

| | property-pair | description | #instances | #premises / prompt |
|---|---|---|---|---|
| ○ | P6_P19 | head of government birth place | 7299 | 41.14 |
| × | P286_P17 | head coach country | 1607 | 17.50 |
| ⊖ | P159_P206 | headquarter location in or next to body of water | 147 | 12.03 |
| ⋆ | P7779_Q21506450 | military unit size class | 90 | 8.21 |
| □ | P8047_P30 | ship country of registry continent | 74 | 6.37 |

Table 2: Property-pairs with their marks in Figure 2 and descriptions, the number of instances (each yielding a *de dicto* and *de re* prompt), and average number of premises per prompt.

**Avoiding event-referencing properties.** We also avoid referencing events in the property ascription, which can cause 'in MONTHYEAR' to be read as referring to the time of the event, for example:

> *"In MONTHYEAR, the club with X **as its head coach was founded by** Y."*

Sometimes we can avoid this by rephrasing:

> *"In MONTHYEAR, the club with X as its head coach had Y **as its** founder."*

One property-pair was discarded because we could not find a good way to phrase the sentence, this was about the location of formation (P740) of distributors of creative works (P750).

## 5 Experiments

We evaluate three dialogue-oriented large language models (LLMs) on our benchmark in a zero-shot setup. The models we include are:

(a) Koala-13B[†] (Geng et al., 2023)[1]

(b) LLaMA-33B[†] (Touvron et al., 2023) finetuned on version 7 of OpenAssistant Conversations (Köpf et al., 2023)[2]

(c) GPT-3.5(-turbo-0301) (OpenAI, 2022)

(d) GPT-4(-0613) (OpenAI, 2022)

Models marked with † were used with 4-bit OPTQ quantization (Frantar et al., 2023).

The evaluation is performed on 500 instances (100 per property-pair). To prompt the language models, we start with instances such as the one from the 'example' column in Table 1 and number the premises P1 through P$N$. Next, we shuffle the order of the interpretations, and then append one of the following two instructions: direct=*"Answer only with "Option 1" or "Option 2", explain your decision after."* or chain-of-thought=*"When answering, let's consider both options, and think step by step. DO NOT repeat the premises, only refer to their number."* When using chain-of-thought (CoT), we follow up with a second message: *"Based on this, what is your final answer,*

[1] hf.co/TheBloke/koala-13B-GPTQ-4bit-128g
[2] hf.co/TheBloke/OpenAssistant-SFT-7-Llama-30B-GPTQ

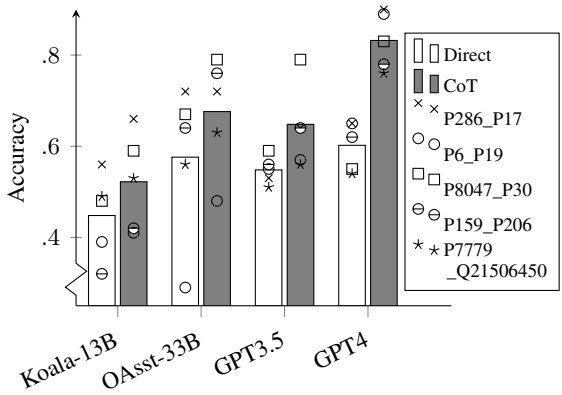

Figure 2: Accuracies obtained by each model on the RADD-Wikidata-5-EN benchmark. Bars show average accuracy, marks overlaid on top indicate accuracy on each of the five property-pairs.

*"Option 1" or "Option 2"?"* The models were prone to repeating the premises in their answer, thus warning them not to do this was necessary to avoid them from running out of context-window before providing the final answer.

### 5.1 Quantitative results

Figure 2 shows the accuracies obtained by each model for each property-pair and on average. The precision, recall and F1 scores for each model, class, and prompt style can be seen in Table 3. The highest average accuracy is obtained by GPT4 (83.2%), followed by OAsst-33B (67.7%) and GPT3.5's (64.8%). The mean accuracy of Koala-13B is roughly equal to random guessing (52.2%). Looking at the performance for the various property-pairs, we can see that GPT3.5, although less accurate overall, is more consistent than OAsst-33B. The performance for OAsst-33B is particularly low on the P6_P19 pair. This can be explained by the average number of premises (41.14) which is over twice the amount as included for the other property pairs (6.37 - 17.5 on average, never fewer than 5). The long prompts exceed the number of tokens included in the context window during training for OAsst-33B (and Koala-13B).

|  | de dicto | | | de re | | |
|---|---|---|---|---|---|---|
|  | Pr | Re | F1 | Pr | Re | F1 |
| **Direct** Koala-13B | 0.46 | 0.42 | 0.44 | 0.47 | *0.47* | *0.47* |
| OAsst-33B | *0.59* | 0.79 | 0.67 | 0.61 | 0.36 | 0.45 |
| GPT3.5 | 0.53 | **1.00** | 0.69 | 0.97 | 0.10 | 0.18 |
| GPT4 | 0.56 | **1.00** | *0.72* | **1.00** | 0.20 | 0.33 |
| **CoT** Koala-13B | 0.56 | 0.51 | 0.53 | 0.56 | 0.54 | 0.54 |
| OAsst-33B | 0.65 | 0.84 | 0.73 | 0.80 | 0.51 | 0.61 |
| GPT3.5 | 0.64 | 0.82 | 0.72 | 0.73 | 0.48 | 0.56 |
| GPT4 | **0.76** | *0.99* | **0.86** | *0.99* | **0.67** | **0.79** |

Table 3: Precision, recall and F1 score by class (columns), for each prompt styles and model (rows). Bold and italic show best scores overall and per prompt-style respectively.

|  |  | Direct | | | CoT | | |
|---|---|---|---|---|---|---|---|
|  |  | de dicto | de re | neither | de dicto | de re | neither |
| Koala-13B | de dicto | 21.2 | 27.2 | 1.6 | 25.4 | 20.6 | 4.0 |
|  | de re | 25.0 | 23.6 | 1.4 | 19.8 | 26.8 | 3.4 |
| OAsst-33B | de dicto | 39.6 | 6.2 | 4.2 | 42.2 | 6.0 | 1.8 |
|  | de re | 27.6 | 18.0 | 4.4 | 22.8 | 25.4 | 1.8 |
| GPT3.5 | de dicto | 49.8 | 0.2 | 0.0 | 41.0 | 8.8 | 0.2 |
|  | de re | 45.0 | 5.0 | 0.0 | 23.8 | 23.8 | 2.4 |
| GPT4 | de dicto | 50.0 | 0.0 | 0.0 | 49.6 | 0.4 | 0.0 |
|  | de re | 39.8 | 10.2 | 0.0 | 16.2 | 33.6 | 0.2 |

Table 4: Average confusion. The columns show prompt styles and predicted classes, the rows show models and label classes. Correct predictions are in gray.

Note also that each model benefits from the use of chain-of-thought prompting.

In Table 4, we can see the confusion table for each model and prompt style. We can see that there are (up to 4%) cases where the models do not produce a parseable result. Finally, we observe that the *de re* instances are the more difficult class; only GPT4 with chain-of-thought is able to perform better than random guessing on these instances.

### 5.2 Error Analysis

We perform a small analysis of the answers given by the best-performing model GPT4 to give insight into what areas need the most improvement.

About half of the mistakes made by GPT4 involve the model not properly following the chain-of-thought instruction, meaning the model already made a prediction in the first one or two sentences of its response. Although, ignoring this instruction seems to be common among both the *de dicto* and *de re* classes. It seems that absent any reasoning the

models have a strong preference for predicting *de dicto* (this can also be seen from the confusion matrix in Table 4). This explains why this behaviour mostly produces errors on the *de re* class.

About a third of all *de re* chain-of-thought answers conclude that neither option is correct. Although despite of this GPT4 almost always still predicts one of the two options. This seems to indicate that the models completely fail to consider the *de re* option. Take for example the following ambiguous sentence *"The birth place of Prague 6's head of state was Třebíč in November 2001"* for which the correct interpretation is Option 2: 'Prague 6's current head of state (Marie Kousalíková) was, as of November 2001, born in Třebíč'. GPT4 reasons as follows: "*Option 2 can be eliminated based on premises (P6) and (P9), as Marie Kousalíková was born in Třebíč, **but she was not the head of state in November 2001**, according to (P1) and (P2). Hence, this interpretation would be incorrect.*" But, this reasoning is clearly incorrect, since her being head of state in November 2001 is not required for Option 2 to be correct. This type of error seems to be a large part of why the models perform worse for *de re* instances.

Together these two types of errors appear to make up the vast majority of GPT4's mistakes.

## 6 Conclusion

We have introduced Reasoning about Ambiguous Definite Descriptions as a way to evaluate how well systems can use natural language reasoning to resolve ambiguous language and have created the first benchmark dataset to demonstrate the value of this task. Our findings show that recent LLMs are not yet capable of solving this task reliably, and that they particularly struggle with *de re* instances. Our error analysis suggests that models—besides not always 'thinking step-by-step' as instructed—do not properly consider the *de re* options, instead hallucinating extra conditions for their correctness.

In future work, we will expand our work by: (1) testing how changing the provided information (e.g. removing regularities) affects the performance; (2) embedding this problem in other tasks such that the ability to solve it depends on the resolution of the ambiguity; (3) constructing multi-lingual versions to evaluate if the ability to resolve ambiguity through reasoning is independent of the language; and (4) extending our method to other extensional operators, and other types of ambiguities.

## Limitations

The *de dicto* / *de re* ambiguity we use in our dataset is one of many possible kinds of ambiguities. Completely excluding the possibility that our results rely on the particulars of this ambiguity will require a diverse set of ambiguities.

So far we have only included English prompts in our benchmark. We leave the creation of prompt templates in other languages for future work.

We have performed a 'best-effort' tuning of the prompts. It is plausible that with a more extensive tuning better prompts can be found and that overall performance could be improved to some degree. An extensive tuning may also reveal that each model benefits from different prompts, which could also change their relative performance.

## Acknowledgements

This research was supported by Huawei Finland through the DreamsLab project. All content represented the opinions of the authors, which were not necessarily shared or endorsed by their respective employers and/ or sponsors.

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
