# OpenReview forum: "Reasoning about Ambiguous Definite Descriptions"
_EMNLP/2023/Conference — EMNLP 2023 Findings_

### Official Review · Reviewer_xFyH · 2023-07-30

**Soundness:** 3

**Excitement:**

2: Mediocre: This paper makes marginal contributions (vs non-contemporaneous work), so I would rather not see it in the conference.

**Paper Topic And Main Contributions:**

The paper introduces a way to evaluate how well systems can use natural language reasoning to resolve ambiguous determinate descriptions.
A benchmark dataset has been created to experiment on this task, based primarily on descriptions that have different referents through time. Examples are produced by means of templates and therefore they might be fairly repetitive. A fine-tuned system might become good at solving the challenge.
Experimental on the benchmark are performed in a zero-shot setting, with no attempt to optimise the prompt to each model, which is known in other tasks may have significant effect on performance.
The best performance of the three tested models is an accuracy of 67.7%, but it can be hardly considered conclusive.

**Reasons To Accept:**

The question about large language models’ ability to solve complex language understanding tasks, is an important one. The paper examines though one particular task.
The dataset is released publicly.

**Reasons To Reject:**

The paper only addresses one particular task of NLU, particularly challenging because it involves temporal reasoning, on which LLMs are known to struggle.
The task is quite hard also for humans, hence a human performance baseline should be included in the study.
The conclusion from the experiments are hard to generalise.
The experiments do not consider fine-tuning the models on the task: since the dataset is semi-automatically constructed, it would be interesting to see how a model trained on the dataset might pick-up the knowledge necessary to tackle the task.

**Reproducibility:**

4: Could mostly reproduce the results, but there may be some variation because of sample variance or minor variations in their interpretation of the protocol or method.

**Reviewer Confidence:**

4: Quite sure. I tried to check the important points carefully. It's unlikely, though conceivable, that I missed something that should affect my ratings.

---

> ### Author Rebuttal · Authors · 2023-08-28
>
> Thank you for the review!
>
> We agree that a human baseline would be valuable, but we think that measuring the relative performance of LLMs is valuable independently of human performance. Therefore, we believe that our current results form a sufficient contribution.
>
> Both prompt engineering and fine-tuning seek to find small changes to the conditions of evaluation that work in a model’s favor. The reason we did not do these experiments is because we are primarily interested in evaluating the *emergent* reasoning capabilities of LLMs. The prompt-tuning we did was aimed at trying to find a clear formulation of the problem that permits each LLM to do reasonably well. Extensive prompt engineering would not benefit our evaluation, because a model that does well at this task should do so regardless of the way the prompt is phrased as long as it is semantically equivalent.
> With regards to fine-tuning, we wanted to avoid the model relying on some unknown biases in our data, rather than its emergent reasoning abilities. While we could have attempted to eliminate as many of these biases as possible, to know for sure if we were actually learning to reason we would have had to evaluate on other reasoning tasks as well. Such experiments would have required more research and a considerably longer paper.
>
> In summary, in our work we focus on creating an *evaluative* benchmark, and measure how well emergent reasoning generalizes to our new task. We will further emphasize this goal in our paper. We leave a human baseline and the creation of training resources for future work.

---

### Official Review · Reviewer_VEf9 · 2023-08-02

**Soundness:** 4

**Excitement:**

4: Strong: This paper deepens the understanding of some phenomenon or lowers the barriers to an existing research direction.

**Paper Topic And Main Contributions:**

This paper introduces a new dataset and benchmark for evaluating the ability of models to resolve ambiguity in language through reasoning. The key idea is using ambiguous definite descriptions, where a phrase like "the pope" could refer to two different entities based on a temporal qualifier. By providing contextual facts, one interpretation can be excluded through reasoning.
The authors designed the prompts to supply all necessary information to resolve the ambiguity, requiring reasoning without additional knowledge. Experiments with recent LLMs like ChatGPT showed low performance, indicating this task is challenging.

**Questions For The Authors:**

Question A: Were there other types of ambiguities you considered using?

Question B: Have you done any deeper analysis into the types of reasoning failures the models exhibit? Are there certain logical connectives or steps that are more problematic?

Question C: You show *de re* problems are harder across models. Do you have any hypotheses for why this is the case?

**Reasons To Accept:**

- Addresses an interesting, underexplored problem - ambiguity resolution through reasoning.
- New semi-automatically generated benchmark dataset based on real-world knowledge.
- Careful prompt design that isolates reasoning ability
- Solid experiments across major recent dialogue models showing poor performance.
- The paper is clear and well written

**Reasons To Reject:**

- There are opportunities to strengthen analysis and gain more insights:
	- what types of reasoning failures occur?
	- Finer-grained breakdown (per instance or per property)
- Discussion could be expanded with more concrete future work proposals based on insights gained.

However, I understand that, due to page limitations, these remarks can be addressed in the next iteration of the present work.

**Reproducibility:**

5: Could easily reproduce the results.

**Reviewer Confidence:**

3: Pretty sure, but there's a chance I missed something. Although I have a good feel for this area in general, I did not carefully check the paper's details, e.g., the math, experimental design, or novelty.

**Typos Grammar Style And Presentation Improvements:**

- line 73: add a comma before "we"
- line 181-182: add commas before and after "by changing property we ascribe to the denotee"

---

> ### Author Rebuttal · Authors · 2023-08-28
>
> Thank you for the review!
>
> We agree that there are additional opportunities for insight. We will add a qualitative analysis of the type of reasoning failures that occur.
> And we do indeed have more ideas for future work that were previously cut to meet the page limit. We will add these back.
>
>
> **Question A**
> While the original idea for this research focused only on de dicto / de re we have thought about ways to expand it to other types of ambiguity.
> In the context of analyzing reasoning about toxic language we have considered examples with other types of ambiguity, for example:
> ```
> From 2016 to 2019 the prime minister of the UK was Theresa May.
> From 2019 to 2022 the prime minister of the UK was Boris Johnson.
> Theresa May is a woman.
> Boris Johnson is a man.
> If the year is [2017/2020] and we are in the UK, does the following sentence express a stereotype?
> “Why is the prime minister always so hysterical?”
> ```
> which relies on the lexical ambiguity of ‘hysterical’.
>
> **Question B**
> A deeper analysis of the failure modes of LLM reasoning is very interesting to us. A quantitative analysis of this will be the subject of future work.
>
> **Question C**
> One possibility is that when these ambiguities occur, in the training data or in English in general, de dicto is more likely to be the intended interpretation. If this is the case it is possible that LLMs pick up on it. We are not aware of research that speaks to the prior probability of either interpretation.
>
>
> We agree with the grammar / style improvements and will make the suggested changes.

---

### Official Review · Reviewer_sY4f · 2023-08-05

**Soundness:** 4

**Excitement:**

4: Strong: This paper deepens the understanding of some phenomenon or lowers the barriers to an existing research direction.

**Paper Topic And Main Contributions:**

This paper investigates whether large language models can distinguish between de dicto and de re interpretations of definite descriptions in a temporal context. The authors generate a dataset, RADD-Wikidata-5-EN, inserting data from 5 English Wikidata property-pairs into context and ambiguous sentence templates, such that the property being ascribed to the denotee determines the correct interpretation of the ambiguous sentence. They test three large language models using both direct and chain-of-thought prompts, finding that the models generally struggle with this task, especially with de re interpretations, although chain-of-thought prompting helps somewhat.

**Questions For The Authors:**

Question A: The authors state, regarding avoiding event-referencing properties, “If we cannot find a good way to phrase it, we discard the property all together” (lines 221-222). How many properties were discarded for this reason?

Question B: The chain-of-thought prompt included the instruction “DO NOT repeat the premises, only refer to their number” (lines 244-245). What was the rationale for this?

**Reasons To Accept:**

This paper describes an interesting twist on the idea of testing whether language models can resolve ambiguity in language—instead of focusing on implicit reasoning based on world knowledge, it studies explicit logical reasoning about definite descriptions. The methods are mostly sound, and the paper is well-written overall. I appreciate that the code and data are publicly available, which should help others reproduce the results.

**Reasons To Reject:**

The dataset contains, and the evaluation is performed on, only five property-pairs, which may introduce variability in the results. While this is probably sufficient to demonstrate the authors’ main claim, that recent large language models struggle with this task, other points, such as the relative performance of OAsst-33B and ChatGPT, may have been different had a different set of property-pairs been chosen.

**Reproducibility:**

5: Could easily reproduce the results.

**Reviewer Confidence:**

3: Pretty sure, but there's a chance I missed something. Although I have a good feel for this area in general, I did not carefully check the paper's details, e.g., the math, experimental design, or novelty.

**Typos Grammar Style And Presentation Improvements:**

The authors may want to consider presenting (in addition to accuracy and the confusion table) the F1 score for each class—this can succinctly show the difference in model performance classifying de re and de dicto instances.

---

> ### Author Rebuttal · Authors · 2023-08-28
>
> Thank you for the review!
>
> We recognize that more property-pairs is better, but rather than constructing as many pairs as possible (with many being similar), we prioritized finding a few representative property-pairs that are quite different. Of course, we specifically required a property whose value changes over time, and a property whose value does not. Furthermore, the combination of properties also must yield sufficient results (we required at least 50 to create the 100 samples). Finally, we also do not want those 50 to be dominated by a single entity, for example, had the data for the ‘person-military_unit-size_class’ pair consisted of only a handful of people changing units many different times, we would not have included it. We will expand the description of this process in the paper, so that larger versions of this benchmark may be easily created in the future.
>
>
> **Question A**
> The only template we couldn’t phrase in a way we were happy with was about the location of formation (`P740`) of distributors of creative works (`P750`).
>
> **Question B**
> We noticed that the LLMs were prone to repeating the information before actually answering the question. This made the answers of the LLM considerably longer, and increased the number of times that the responses exceeded the context-window. Since repeating the answers does not really add any additional information, we tried to discourage this behavior by adding this instruction. We will clarify this in the paper.
>
>
> We agree that F1-score for each class is a good addition to better show the difference between classes, we will add this to our results section.

---

### Meta-Review · Area_Chair_L1EU · 2023-09-17

**Recommendation:** 3

**Metareview:**

This paper introduces a dataset which requires models to resolve ambiguous definite descriptions (phrases that denote entities by describing the properties or roles that are unique to them within the relevant context, e.g. "the pope", "john’s mother", "our king"). The dataset has been constructed semi-automatically using 5 property-value pairs from from Wikidata. Their experiments indicate that LLMs still struggle to resolve this kind of ambiguity.

The paper tackles an interesting problem and makes a valuable contribution in terms of the dataset. It does have drawbacks -- there's no human baseline, the dataset has been constructed from only 5 property value pairs, and could have used stronger baselines (even if as upper bounds). However, I think the paper meets the criteria for a short paper, i.e., a short focused contribution which the community can build upon.

---

### Decision · Program_Chairs · 2023-10-07

**Decision:**

Accept-Findings

**Comment:**

This paper introduces a dataset which requires models to resolve ambiguous definite descriptions (phrases that denote entities by describing the properties or roles that are unique to them within the relevant context, e.g. "the pope", "john’s mother", "our king"). The dataset has been constructed semi-automatically using 5 property-value pairs from from Wikidata. Their experiments indicate that LLMs still struggle to resolve this kind of ambiguity.

The paper tackles an interesting problem and makes a valuable contribution in terms of the dataset. It does have drawbacks -- there's no human baseline, the dataset has been constructed from only 5 property value pairs, and could have used stronger baselines (even if as upper bounds). However, I think the paper meets the criteria for a short paper, i.e., a short focused contribution which the community can build upon.